# Discriminative State-Space Models

**Vitaly Kuznetsov**
Google Research
New York, NY 10011, USA
vitaly@cims.nyu.edu

**Mehryar Mohri**
Courant Institute and Google Research
New York, NY 10011, USA
mohri@cims.nyu.edu

## Abstract

We introduce and analyze Discriminative State-Space Models for forecasting non-stationary time series. We provide data-dependent generalization guarantees for learning these models based on the recently introduced notion of discrepancy. We provide an in-depth analysis of the complexity of such models. We also study the generalization guarantees for several structural risk minimization approaches to this problem and provide an efficient implementation for one of them which is based on a convex objective.

## 1   Introduction

Time series data is ubiquitous in many domains including such diverse areas as finance, economics, climate science, healthcare, transportation and online advertisement. The field of time series analysis consists of many different problems, ranging from analysis to classification, anomaly detection, and forecasting. In this work, we focus on the problem of forecasting, which is probably one of the most challenging and important problems in the field.

Traditionally, time series analysis and time series prediction, in particular, have been approached from the perspective of generative modeling: particular generative parametric model is postulated that is assumed to generate the observations and these observations are then used to estimate unknown parameters of the model. Autoregressive models are among the most commonly used types of generative models for time series [Engle, 1982, Bollerslev, 1986, Brockwell and Davis, 1986, Box and Jenkins, 1990, Hamilton, 1994]. These models typically assume that the stochastic process that generates the data is stationary up to some known transformation, such as differencing or composition with natural logarithms.

In many modern real world applications, the stationarity assumption does not hold, which has led to the development of more flexible generative models that can account for non-stationarity in the underlying stochastic process. *State-Space Models* [Durbin and Koopman, 2012, Commandeur and Koopman, 2007, Kalman, 1960] provide a flexible framework that captures many of such generative models as special cases, including autoregressive models, hidden Markov models, Gaussian linear dynamical systems and many other models. This framework typically assumes that the time series $\mathbf{Y}$ is a noisy observation of some dynamical system $\mathbf{S}$ that is hidden from the practitioner:

$$Y_t = h(S_t) + \epsilon_t, \quad S_t = g(S_{t-1}) + \eta_t \quad \text{for all } t. \tag{1}$$

In (1), $h$, $g$ are some unknown functions estimated from data, $\{\epsilon_t\}$, $\{\eta_t\}$ are sequences of random variables and $\{S_t\}$ is an unobserved sequence of states of a hidden dynamical system.[1] While this class of models provides a powerful and flexible framework for time series analysis, the theoretical learning properties of these models is not sufficiently well understood. The statistical guarantees available in the literature rely on strong assumptions about the noise terms (e.g. $\{\epsilon_t\}$ and $\{\eta_t\}$ are Gaussian white noise). Furthermore, these results are typically asymptotic and require the model

to be correctly specified. This last requirement places a significant burden on a practitioner since the choice of the hidden state-space is often a challenging problem and typically requires extensive domain knowledge.

In this work, we introduce and study *Discriminative State-Space Models (DSSMs)*. We provide the precise mathematical definition of this class of models in Section 2. Roughly speaking, a DSSM follows the same general structure as in (1) and consists of a state predictor $g$ and an observation predictor $h$. However, no assumption is made about the form of the stochastic process used to generate observations. This family of models includes existing generative models and other state-based discriminative models (e.g. RNNs) as special cases, but also consists of some novel algorithmic solutions explored in this paper.

The material we present is organized as follows. In Section 3, we generalize the notion of discrepancy, recently introduced by Kuznetsov and Mohri [2015] to derive learning guarantees for DSSMs. We show that our results can be viewed as a generalization of those of these authors. Our notion of discrepancy is finer, taking into account the structure of state-space representations, and leads to tighter learning guarantees. Additionally, our results provide the first high-probability generalization guarantees for state-space models with possibly incorrectly specified models. Structural Risk Minimization (SRM) for DSSMs is analyzed in Section 4. As mentioned above, the choice of the state-space representation is a challenging problem since it requires carefully balancing the accuracy of the model on the training sample with the complexity of DSSM to avoid overfitting. We show that it is possible to adaptively learn a state-space representation in a principled manner using the SRM technique. This requires analyzing the complexity of several families of DSSMs of interest in Appendix B. In Section 5, we use our theory to design an efficient implementation of our SRM technique. Remarkably, the resulting optimization problem turns out to be convex. This should be contrasted with traditional SSMs that are often derived via Maximum Likelihood Estimation (MLE) with a non-convex objective. We conclude with some promising preliminary experimental results in Appendix D.

## 2   Preliminaries

In this section, we introduce the general scenario of time series prediction as well as the broad family of DSSMs considered in this paper.

We study the problem of time series forecasting in which the learner observes a realization $(X_1, Y_1), \ldots, (X_T, Y_T)$ of some stochastic process, with $(X_t, Y_t) \in \mathcal{Z} = \mathcal{X} \times \mathcal{Y}$. We assume that the learner has access to a family of observation predictors $\mathcal{H} = \{h \colon \mathcal{X} \times \mathcal{S} \to \mathcal{Y}\}$ and state predictors $\mathcal{G} = \{g \colon \mathcal{X} \times \mathcal{S} \to \mathcal{S}\}$, where $\mathcal{S}$ is some pre-defined space. We refer to any pair $f = (h, g) \in \mathcal{H} \times \mathcal{G} = \mathcal{F}$ as a DSSM, which is used to make predictions as follows:

$$y_t = h(X_t, s_t), \quad s_t = g(X_t, s_{t-1}) \quad \text{for all } t. \tag{2}$$

Observe that this formulation includes the hypothesis sets used in (1) as special cases. In our setting, $h$ and $g$ both accept an additional argument $x \in \mathcal{X}$. In practice, if $X_t = (Y_{t-1}, \ldots, Y_{t-p}) \in \mathcal{X} = \mathcal{Y}^p$ for some $p$, then $X_t$ represents some recent history of the stochastic process that is used to make a prediction of $Y_t$. More generally, $\mathcal{X}$ may also contain some additional side information. Elements of the output space $\mathcal{Y}$ may further be multi-dimensional, which covers both multi-variate time series forecasting and multi-step forecasting.

The performance of the learner is measured using a bounded loss function $L \colon \mathcal{H} \times \mathcal{S} \times \mathcal{Z} \to [0, M]$, for some upper bound $M \geq 0$. A commonly used loss function is the squared loss: $L(h, s, z) = (h(x, s) - y)^2$.

The objective of the learner is to use the observed realization of the stochastic process up to time $T$ to determine a DSSM $f = (h, g) \in \mathcal{F}$ that has the smallest expected loss at time $T + 1$, conditioned on the given realization of the stochastic process:[2]

$$\mathcal{L}_{T+1}(f | \mathbf{Z}_1^T) = \mathbb{E}[L(h, s_{T+1}, Z_{T+1}) | \mathbf{Z}_1^T], \tag{3}$$

where $s_t$ for all $t$ is specified by $g$ via the recursive computation in (2). We will use the notation $\mathbf{a}_s^r$ to denote $(a_s, a_{s+1}, \ldots a_r)$.

In the rest of this section, we will introduce the tools needed for the analysis of this problem. The key technical tool that we require is the notion of *state-space discrepancy*:

$$\mathrm{disc}(\mathbf{s}) = \sup_{h \in \mathcal{H}} \left( \mathbb{E}[L(h, s_{T+1}, Z_{T+1})|\mathbf{Z}_1^T] - \frac{1}{T} \sum_{t=1}^T \mathbb{E}[L(h, s_t, Z_t)|\mathbf{Z}_1^{t-1}] \right), \quad (4)$$

where, for simplicity, we used the shorthand $\mathbf{s} = \mathbf{s}_1^{T+1}$. This definition is a strict generalization of the **q**-weighted discrepancy of Kuznetsov and Mohri [2015]. In particular, redefining $L(h, s, z) = s\widetilde{L}(h, z)$ and setting $s_t = Tq_t$ for $1 \le t \le T$ and $s_{T+1} = 1$ recovers the definition of **q**-weighted discrepancy. The discrepancy $\mathrm{disc}$ defines an integral probability pseudo-metric on the space of probability distributions that serves as a measure of the non-stationarity of the stochastic process $\mathbf{Z}$ with respect to both the loss function $L$ and the hypothesis set $\mathcal{H}$, conditioned on the given state sequence $\mathbf{s}$. For example, if the process $\mathbf{Z}$ is i.i.d., then we simply have $\mathrm{disc}(\mathbf{s}) = 0$ provided that $\mathbf{s}$ is a constant sequence. See [Cortes et al., 2017, Kuznetsov and Mohri, 2014, 2017, 2016, Zimin and Lampert, 2017] for further examples and bounds on discrepancy in terms of other divergences. However, the most important property of the discrepancy $\mathrm{disc}(\mathbf{s})$ is that, as shown in Appendix C, under some additional mild assumptions, it can be estimated from data.

The learning guarantees that we present are given in terms of data-dependent measures of sequential complexity, such as expected sequential covering number [Rakhlin et al., 2010], that are modified to account for the state-space structure in the hypothesis set. The following definition of a complete binary tree is used throughout this paper: a $\mathcal{Z}$-valued complete binary tree $\mathbf{z}$ is a sequence $(z_1, \ldots, z_T)$ of $T$ mappings $z_t \colon \{\pm 1\}^{t-1} \to \mathcal{Z}, t \in [1, T]$. A path in the tree is $\sigma = (\sigma_1, \ldots, \sigma_{T-1}) \in \{\pm 1\}^{T-1}$. We write $z_t(\boldsymbol{\sigma})$ instead of $z_t(\sigma_1, \ldots, \sigma_{t-1})$ to simplify the notation. Let $\mathcal{R} = \mathcal{R}' \times \mathcal{G}$ be any function class where $\mathcal{G}$ is a family of state predictors and $\mathcal{R}' = \{r \colon \mathcal{Z} \times \mathcal{S} \to \mathbb{R}\}$. A set $V$ of $\mathbb{R}$-valued trees of depth $T$ is a *sequential $\alpha$-cover* (with respect to $\ell_p$ norm) of $\mathcal{R}$ on a tree $\mathbf{z}$ of depth $T$ if for all $(r, g) \in \mathcal{R}$ and all $\boldsymbol{\sigma} \in \{\pm 1\}^T$, there is $\mathbf{v} \in V$ such that

$$\left[ \frac{1}{T} \sum_{t=1}^T \left| \mathbf{v}_t(\boldsymbol{\sigma}) - r(\mathbf{z}_t(\boldsymbol{\sigma}), s_t) \right|^p \right]^{\frac{1}{p}} \le \alpha,$$

where $s_t = g(\mathbf{z}_t(\boldsymbol{\sigma}), s_{t-1})$. The *(sequential) covering number* $\mathcal{N}_p(\alpha, \mathcal{R}, \mathbf{z})$ on a given tree $\mathbf{z}$ is defined to be the size of the minimal sequential cover. We call $\mathcal{N}_p(\alpha, \mathcal{R}) = \sup_{\mathbf{z}} \mathcal{N}_p(\alpha, \mathcal{R}, \mathbf{z})$ the *maximal covering number*. See Figure 1 for an example.

We define the *expected covering number* to be $\mathbb{E}_{\mathbf{z} \sim T(\mathbf{p})}[\mathcal{N}_p(\alpha, \mathcal{R}, \mathbf{z})]$, where $T(\mathbf{p})$ denotes the distribution of $\mathbf{z}$ implicitly defined via the following sampling procedure. Given a stochastic process distributed according to the distribution $\mathbf{p}$ with $\mathbf{p}_t(\cdot|\mathbf{z}_1^{t-1})$ denoting the conditional distribution at time $t$, sample $Z_1, Z_1'$ from $\mathbf{p}_1$ independently. In the left child of the root sample $Z_2, Z_2'$ according to $\mathbf{p}_2(\cdot|Z_1)$ and in the right child according to $\mathbf{p}_2(\cdot|Z_2')$ all independent from each other. For a node that can be reached by a path $(\sigma_1, \ldots, \sigma_t)$, we draw $Z_t, Z_t'$ according to $\mathbf{p}_t(\cdot|S_1(\sigma_1), \ldots, S_{t-1}(\sigma_{t-1}))$, where $S_t(1) = Z_t$ and $S_t(-1) = Z_t'$. Expected sequential covering numbers are a finer measure of complexity since they directly take into account the distribution of the underlying stochastic process.

For further details on sequential complexity measures, we refer the reader to [Littlestone, 1987, Rakhlin et al., 2010, 2011, 2015a,b].

# 3 Theory

In this section, we present our generalization bounds for learning with DSSMs. For our first result, we assume that the sequence of states $\mathbf{s}$ (or equivalently state predictor $g$) is fixed and we are only learning the observation predictor $h$.

**Theorem 1.** *Fix $\mathbf{s} \in \mathcal{S}^{T+1}$. For any $\delta > 0$, with probability at least $1 - \delta$, for all $h \in \mathcal{H}$ and all $\alpha > 0$, the following inequality holds:*

$$\mathcal{L}(f|\mathbf{Z}_1^T) \le \frac{1}{T} \sum_{t=1}^T L(h, X_t, s_t) + \mathrm{disc}(\mathbf{s}) + 2\alpha + M \sqrt{\frac{2 \log \frac{\mathbb{E}_{\mathbf{v} \sim T(\mathbb{P})}[\mathcal{N}_1(\alpha, \mathcal{R}_{\mathbf{s}}, \mathbf{v})]}{\delta}}{T}},$$

where $\mathcal{R}_{\mathbf{s}} = \{(z,s) \mapsto L(h,s,z) \colon h \in \mathcal{H}\} \times \{\mathbf{s}\}$.

The proof of Theorem 1 (as well as the proofs of all other results in this paper) is given in Appendix A. Note that this result is a generalization of the learning guarantees of Kuznetsov and Mohri [2015]. Indeed, setting $\mathbf{s} = (Tq_1, \ldots, Tq_T, 1)$ for some weight vector $\mathbf{q}$ and $L(h,s,z) = s\widetilde{L}(h,z)$ recovers Corollary 2 of Kuznetsov and Mohri [2015]. Zimin and Lampert [2017] show that, under some additional assumptions on the underlying stochastic process (e.g. Markov processes, uniform martingales), it is possible to choose these weights to guarantee that the discrepancy $\mathrm{disc}(\mathbf{s})$ is small. Alternatively, Kuznetsov and Mohri [2015] show that if the distribution of the stochastic process at times $T+1$ and $[T-s,T]$ is sufficiently close (in terms of discrepancy) then $\mathrm{disc}(\mathbf{s})$ can be estimated from data. In Theorem 5 in Appendix C, we show that this property holds for arbitrary state sequences $\mathbf{s}$. Therefore, one can use the bound of Theorem 1 that can be computed from data to search for the predictor $h \in \mathcal{H}$ that minimizes this quantity. The quality of the result will depend on the given state-space sequence $\mathbf{s}$. Our next result shows that it is possible to learn $h \in \mathcal{H}$ and $\mathbf{s}$ generated by some state predictor $g \in \mathcal{G}$ jointly.

**Theorem 2.** *For any $\delta > 0$, with probability at least $1 - \delta$, for all $f = (h,g) \in \mathcal{H} \times \mathcal{G}$ and all $\alpha > 0$, the following inequality holds:*

$$\mathcal{L}(f|\mathbf{Z}_1^T) \le \frac{1}{T}\sum_{t=1}^{T} L(h,X_t,s_t) + \mathrm{disc}(\mathbf{s}) + 2\alpha + M\sqrt{\frac{2\log\frac{\mathbb{E}_{\mathbf{v} \sim T(\mathbb{P})}[\mathcal{N}_1(\alpha,\mathcal{R},\mathbf{v})]}{\delta}}{T}},$$

*where $s_t = g(X_t, s_{t-1})$ for all $t$ and $\mathcal{R} = \{(z,s) \mapsto L(h,s,z) \colon h \in \mathcal{H}\} \times \mathcal{G}$.*

The cost of this significantly more general result is a slightly larger complexity term $\mathcal{N}_1(\alpha,\mathcal{R},\mathbf{v}) \ge \mathcal{N}_1(\alpha,\mathcal{R}_{\mathbf{s}},\mathbf{v})$. This bound is also much tighter than the one that can be obtained by applying the result of Kuznetsov and Mohri [2015] directly to $\mathcal{F} = \mathcal{H} \times \mathcal{G}$, which would lead to the same bound as in Theorem 2 but with $\mathrm{disc}(\mathbf{s})$ replaced by $\sup_{g \in \mathcal{G}} \mathrm{disc}(\mathbf{s})$. Not only $\sup_{g \in \mathcal{G}} \mathrm{disc}(\mathbf{s})$ is an upper bound on $\mathrm{disc}(\mathbf{s})$, but it is possible to construct examples that lead to learning bounds that are too loose. Consider the stochastic process generated as follows. Let $X$ be uniformly distributed on $\{\pm 1\}$. Suppose $Y_1 = 1$ and $Y_t = -Y_{t-1}$ for all $t > 1$ if $X = -1$ and $Y_t = Y_{t-1}$ for all $t > 1$ otherwise. In other words, $\mathbf{Y}$ is either periodic or a constant state sequence. If $L$ is the squared loss, for $\mathcal{G} = \{g_1, g_2\}$ with $g_1(s) = s$ and $g_2(s) = -s$ and $\mathcal{H} = \{h\}$ with $h(s) = s$, for odd $T$, $\sup_{g \in \mathcal{G}} \mathrm{disc}(\mathbf{s}) \ge 1/2$. On the other hand, the bound in terms of $\mathrm{disc}(\mathbf{s})$ is much finer and helps us select $g$ such that $\mathrm{disc}(\mathbf{s}) = 0$ for that $g$. This example shows that even for simple deterministic dynamics our learning bounds are finer than existing ones.

Since the guarantees of Theorem 2 are data-dependent and hold uniformly over $\mathcal{F}$, they allow us to seek a solution $f \in \mathcal{F}$ that would directly optimize this bound and that could be computed from the given sample. As our earlier example shows, the choice of the family of state predictors $\mathcal{G}$ is crucial to achieve good guarantees. For instance, if $\mathcal{G} = \{g_1\}$ then it may be impossible to have a non-trivial bound. In other words, if the family of state predictors is not rich enough, then, it may not be possible to handle the non-stationarity of the data. On the other hand, if $\mathcal{G}$ is chosen to be too large, then, the complexity term may be too large. In Section 4, we present an SRM technique that enables us to learn the state-space representation and adapt to non-stationarity in a principled way.

## 4 Structural Risk Minimization

Suppose we are given a sequence of families of observation predictors $\mathcal{H}_1 \subset \mathcal{H}_2 \subset \cdots \mathcal{H}_n \ldots$ and a sequence of families of state predictors $\mathcal{G}_1 \subset \mathcal{G}_2 \cdots G_n \ldots$ Let $\mathcal{R}_k = \{(s,z) \mapsto L(h,s,z) \colon h \in \mathcal{H}_k\} \times \mathcal{G}_k$ and $\mathcal{R} = \cup_{k=1}^{\infty} \mathcal{R}_k$. Consider the following objective function:

$$F(h,g,k) = \frac{1}{T}\sum_{t=1}^{T} L(h,s_t,Z_t) + \Delta(\mathbf{s}) + B_k + M\sqrt{\frac{\log k}{T}}, \qquad (5)$$

where $\Delta(\mathbf{s})$ is any upper bound on $\mathrm{disc}(\mathbf{s})$ and $B_k$ is any upper bound on $M\sqrt{\frac{2\log\frac{\mathbb{E}_{\mathbf{v} \sim T(\mathbb{P})}[\mathcal{N}_1(\alpha,\mathcal{R}_k,\mathbf{v})]}{\delta}}{T}}$. We are presenting an estimatable upper bound on $\mathrm{disc}(\mathbf{s})$ in Appendix C, which provides one

particular choice for $\Delta(\mathbf{s})$. In Appendix B, we also prove upper bounds on the expected sequential covering numbers for several families of hypothesis. Then, we define the SRM solution as follows:

$$(\widetilde{h}, \widetilde{g}, \widetilde{k}) = \operatorname{argmin}_{h,g \in \mathcal{H}_k \times \mathcal{G}_k, k \geq 1} F(h, g, k). \tag{6}$$

We also define $f^*$ by $f^* = (h^*, g^*) \in \operatorname{argmin}_{f \in \mathcal{F}} \mathcal{L}_{T+1}(f|\mathbf{Z}_1^T)$. Then, the following result holds.

**Theorem 3.** *For any $\delta > 0$, with probability at least $1 - \delta$, for all $\alpha > 0$, the following bound holds:*

$$\mathcal{L}_{T+1}(\widetilde{h}, \widetilde{g}|\mathbf{Z}_1^T) \leq \mathcal{L}_{T+1}(f^*|\mathbf{Z}_1^T) + 2\Delta(\mathbf{s}^*) + 2\alpha + 2B_{k(f^*)} + M\sqrt{\frac{\log k(f^*)}{T}} + 2M\sqrt{\frac{\log \frac{2}{\delta}}{T}},$$

*where $s_t^* = g^*(X_t, s_{t-1}^*)$, and where $k(f^*)$ is the smallest integer $k$ such that $f^* \in \mathcal{H}_k \times \mathcal{G}_k$.*

Theorem 3 provides a learning guarantee for the solution of SRM problem (5). This result guarantees for the SRM solution a performance close to that of the best-in-class model $f^*$ modulo a penalty term that includes the discrepancy (of the best-in-class state predictor), similar to the guarantees of Section 3. This guarantee can be viewed as a worst case bound when we are unsure if the state-space predictor captures the non-stationarity of the problem correctly. However, in most cases, by introducing a state-space representation, we hope that it will help us model (at least to some degree) the non-stationarity of the underlying stochastic process. In what follows, we present a more optimistic best-case analysis which shows that, under some additional mild assumptions on the complexity of the hypothesis space with respect to stochastic process, we can simultaneously simplify the SRM optimization and give tighter learning guarantees for this modified version.

**Assumption 1** (Stability of state trajectories). *Assume that there is a decreasing function $r$ such that for any $\epsilon > 0$ and $\delta > 0$, with probability $1 - \delta$, if $h^*, g^* = \operatorname{argmin}_{(h,g) \in \mathcal{F}} \mathcal{L}_{T+1}(h, g|\mathbf{Z}_1^T)$ and $(h, g) \in \mathcal{F}$ is such that*

$$\left| \frac{1}{T} \sum_{t=1}^{T} \mathcal{L}_t(h, g|\mathbf{Z}_1^{t-1}) - \mathcal{L}_t(h^*, g^*|\mathbf{Z}_1^{t-1}) \right| \leq \epsilon, \tag{7}$$

*then, the following holds:*

$$\mathcal{L}_{T+1}(h, g|\mathbf{Z}_1^T) - \mathcal{L}_{T+1}(h^*, g^*|\mathbf{Z}_1^T) \leq r(\epsilon). \tag{8}$$

Roughly speaking, this assumption states that, given a sequence of states $s_1, \ldots, s_T$ generated by $g$ such that the performance of some observation predictor $h$ along this sequence of states is close to the performance of the ideal pair $h^*$ along the ideal sequence generated by $g^*$, the performance of $h$ in the near future (at state $s_{T+1}$) will remain close to that of $h^*$ (in state $s_{T+1}^*$). Note that, in most cases of interest, $r$ has the form $r(\epsilon) = a\epsilon$, for some $a > 0$.

Consider the following optimization problem which is similar to (5) but omits the discrepancy upper bound $\Delta$:

$$F_0(h, g, k) = \frac{1}{T} \sum_{t=1}^{T} L(h, s_t, Z_t) + B_k + M\sqrt{\frac{\log k}{T}}, \tag{9}$$

We will refer to $F_0$ as an *optimistic* SRM objective and we let $(h_0, g_0)$ be a minimizer of $F_0$. Then, we have the following learning guarantee.

**Theorem 4.** *Under Assumption 1, for any $\delta > 0$, with probability at least $1 - \delta$, for all $\alpha > 0$, the inequality $\mathcal{L}_{T+1}(h_0, g_0|\mathbf{Z}_1^T) - \mathcal{L}_{T+1}(f^*|\mathbf{Z}_1^T) < r(\epsilon)$ holds with*

$$\epsilon = 2\alpha + 2B_{k(f^*)} + M\sqrt{\frac{\log k(f^*)}{T}} + 2M\sqrt{\frac{\log \frac{2}{\delta}}{T}},$$

*where $s_t^* = g^*(X_t, s_{t-1}^*)$, and where $k(f^*)$ is the smallest integer $k$ such that $f^* \in \mathcal{H}_k \times \mathcal{G}_k$.*

We remark that a finer analysis can be used to show that Assumption 1 only need to be satisfied for $k \leq k(f^*)$ for the Theorem 4. Furthermore, observe that for linear functions $r(\epsilon) = a\epsilon$, one recovers a guarantee similar to the bound in Theorem 3, but the discrepancy term is omitted making this result tighter. This result suggests that in the optimistic scenarios where our hypothesis set contains a good

state predictor that can capture the data non-stationarity, it is possible to achieve a tighter guarantee that avoids the pessimistic discrepancy term. Note that, increasing the capacity of the family of state predictors makes it easier to find such a good state predictor but it also may make the learning problem harder and lead to the violation of Assumption 1. This further motivates the use of an SRM technique for this problem to find the right balance between capturing the non-stationarity in data and the complexity of the models that are being used. Theorem 4 formalizes this intuition by providing theoretical guarantees for this approach.

We now consider several illustrative examples showing that this assumption holds in a variety of cases of interest. In all our examples, we will use the squared loss but it is possible to generalize all of them to other sufficiently regular losses.

**Linear models.** Let $\mathcal{F}$ be defined by $\mathcal{F} = \{f\colon \mathbf{y} \mapsto \mathbf{w} \cdot \Psi(\mathbf{y}), \|\mathbf{w}\| \leq \Lambda\}$ for some $\Lambda > 0$ and some feature map $\Psi$. Consider a separable case where $Y_t = \mathbf{w}^* \cdot \Psi(\mathbf{Y}_{t-p}^{t-1}) + \epsilon_t$, where $\epsilon_t$ represents white noise. One can verify that the following equality holds:

$$\mathcal{L}_t(\mathbf{w}|\mathbf{Z}_1^{t-1}) = \mathbb{E}[(\mathbf{w} \cdot \Psi(\mathbf{Y}_{t-p}^{t-1}) - Y_t)|\mathbf{Y}_1^{t-1}] = \left[(\mathbf{w} - \mathbf{w}^*) \cdot \Psi(\mathbf{Y}_{t-p}^{t-1})\right]^2.$$

In view of that, it follows that (7) is equal to

$$\frac{1}{T}\sum_{t=1}^{T}\left[(\mathbf{w} - \mathbf{w}^*) \cdot \Psi(\mathbf{Y}_{t-p}^{t-1})\right]^2 \geq \frac{1}{T}\sum_{t=1}^{T}(\mathbf{w}_j - \mathbf{w}_j^*)^2 \Psi_j(\mathbf{Y}_{t-p}^{t-1})^2$$

for any coordinate $j \in [1, N]$. Thus, for any coordinate $j \in [1, N]$, by Hölder's inequality, we have

$$\mathcal{L}_{T+1}(h, g|\mathbf{Z}_1^T) - \mathcal{L}_{T+1}(h^*, g^*|\mathbf{Z}_1^T) = \left[(\mathbf{w} - \mathbf{w}^*) \cdot \Psi(\mathbf{Y}_{T-p+1}^T)\right]^2 \leq r\epsilon\sum_{j=1}^{N}\frac{1}{\sigma_j},$$

where $\sigma_j = \frac{1}{T}\sum_{t=1}^{T}\Psi_j(\mathbf{Y}_{t-p}^{t-1})^2$ is the empirical variance of the $j$-th coordinate and where $r = \sup_{\mathbf{y}} \Psi(\mathbf{y})^2$ is the empirical $\ell_\infty$-norm radius of the data. The special case where $\Psi$ is the identity map covers standard autoregressive models. These often serve as basic building blocks for other state-space models, as discussed below. More generally, other feature maps $\Psi$ may be induced by a positive definite kernel $K$. Alternatively, we may take as our hypothesis set $\mathcal{F}$ the convex hull of all decision trees of certain depth $d$. In that case, we can view each coordinate $\Psi_j$ as the output of a particular decision tree on the given input.

**Linear trend models.** For simplicity, in this example, we consider univariate time series with linear trend. However, this can be easily generalized to the multi-variate setting with different trend models. Define $\mathcal{G}$ as $\mathcal{G} = \{s \mapsto s + c\colon |c| \leq \Lambda\}$ for some $\Lambda > 0$ and let $\mathcal{H}$ be a singleton consisting of the identity map. Assume that $Y_t = c^*t + \epsilon_t$, where $\epsilon_t$ is white noise. As in the previous example, it is easy to check that $\mathcal{L}_t(h, g|\mathbf{Z}_1^{t-1}) = |c - c^*|^2 t^2$. Therefore, in this case, one can show that (7) reduces to $\frac{1}{3}(T+1)(2T+1)|c - c^*|^2$ and therefore, if $\epsilon = O(\sqrt{1/T})$, then we have $|c - c^*|^2 = O(1/T^{5/2})$ and thus (8) is $|c - c^*|^2(T+1)^2 = O(\sqrt{1/T})$.

**Periodic signals.** We study a multi-resolution setting where the time series of interest are modeled as a linear combination of periodic signals at different frequencies. We express this as a state-space model as follows. Define

$$\mathbf{A}_d = \begin{bmatrix} -\mathbf{1} & -1 \\ \mathbf{I}_{d-1} & \mathbf{0} \end{bmatrix},$$

where $\mathbf{1}$ is $d-1$-dimensional row vector of 1s, $\mathbf{0}$ is $d-1$-dimensional column vector of 0 and $\mathbf{I}_{d-1}$ is an identity matrix. It easy to verify that, under the map $\mathbf{s} \mapsto \mathbf{A}_d\mathbf{s}$, the sequence $\mathbf{s}_1 \cdot \mathbf{e}_1, \mathbf{s}_2 \cdot \mathbf{e}_1 \ldots, \mathbf{s}_t \cdot \mathbf{e}_1 \ldots$, where $\boldsymbol{\epsilon}_1 = (1, 0, \ldots, 0)^T$, is a periodic sequence with period $d$. Let $D = d_1, \ldots, d_k$ be any collection of positive integers and let $\mathbf{A}$ be a block-diagonal matrix with $\mathbf{A}_{d_1}, \ldots, \mathbf{A}_{d_k}$ on the diagonal. We set $\mathcal{G} = \{\mathbf{s} \mapsto \mathbf{A} \cdot \mathbf{s}\}$ and $\mathcal{H} = \{\mathbf{s} \mapsto \mathbf{w} \cdot \mathbf{s}\colon \|\mathbf{w}\| < \Lambda\}$, where we also restrict $\mathbf{w}$s to be non-zero only at coordinates $1, 1 + d_1, 1 + d_1 + d_2, \ldots, 1 + \sum_{j=1}^{k-1}d_{k-1}$. Once again, to simplify our presentation, we assume that $Y_t$ satisfies $Y_t = \mathbf{w}^* \cdot \mathbf{s}_t + \epsilon_t$. Using arguments similar to those of the previous examples, one can show that (7) is lower bounded by $(\mathbf{w}_j - \mathbf{w}_j^*)^2\frac{1}{T}\sum_{t=1}^{T}\mathbf{s}_{t,j}$ for any coordinate $j$. Therefore, as before, if (7) is upper bounded by $\epsilon > 0$, then (8) is upper bounded by $r\epsilon\sum_{j=1}^{N}\frac{1}{\sigma_j}$, where $r$ is the maximal radius of any state and $\sigma_j$ a variance of $j$-th state sequence.

**Trajectory ensembles.** Note that, in our previous example, we did not exploit the fact that the sequences were periodic. Indeed, our argument holds for any $g$ that generates a multi-dimensional trajectory $h \in \mathcal{H} = \{\mathbf{s} \mapsto \mathbf{w} \cdot \mathbf{s} \colon \|\mathbf{w}\| < \Lambda\}$ which can be interpreted as learning an ensemble of different state-space trajectories.

**Structural Time Series Models (STSMs).** STSMs are a popular family of state-space models that combine all of the previous examples. For this model, we use $(h, g) \in \mathcal{H} \times \mathcal{G}$ that have the following structure: $h(\mathbf{x}_t, g(\mathbf{s}_t)) = \mathbf{w} \cdot \Psi(\mathbf{x}_t) + ct + \mathbf{w}' \cdot \mathbf{s}_t$, where $\mathbf{s}_t$ is a vector of periodic sequences described in the previous examples and $\mathbf{x}_t$ is the vector representing the most recent history of the time series. Note that our formulation is very general and allows for arbitrary feature maps $\Psi$ that can correspond either to kernel-based or tree-based models. Arguments similar to those given in previous examples show that Assumption 1 holds in this case.

**Shifting parameters.** We consider the non-realizable case where $\mathcal{H}$ is a set of linear models but where the data is generated according to the following procedure. The first $T/2$ rounds obey the formula $Y_t = \mathbf{w}_0 Y_{t-1} + \epsilon_t$, the subsequent rounds the formula $Y_t = \mathbf{w}^* Y_{t-1} + \epsilon_t$. Note that, in this case, we have $|\frac{1}{T} \sum_{t=1}^{T} \mathcal{L}_t(\mathbf{w}_0 | \mathbf{Z}_1^{t-1}) - \mathcal{L}_t(\mathbf{w}^* | \mathbf{Z}_1^{t-1})| = 0$. However, if $\mathbf{w}_0$ and $\mathbf{w}^*$ are sufficiently far apart, it is possible to show that there is a constant lower bound on $\mathcal{L}_{T+1}(\mathbf{w}_0 | \mathbf{Z}_1^T) - \mathcal{L}_{T+1}(\mathbf{w}^* | \mathbf{Z}_1^T)$. One approach to making Assumption 1 hold for this stochastic process is to choose $\mathcal{H}$ such that the resulting learning problem is separable. However, that requires us to know the exact nature of the underlying stochastic process. An alternative agnostic approach, is to consider a sequence of states (or equivalently weights) that can assign different weights $q_t$ to different training points.

Finally, observe that our learning guarantees in Section 3 and 4 are expressed in terms of the expected sequential covering numbers of the family of DSSMs that we are seeking to learn. A priori, it is not clear if it is possible to control the complexity of such models in a meaningful way. However, in Appendix B, we present explicit upper bounds on the expected sequential covering numbers of several families of DSSMs, including several of those discussed above: linear models, tree-based hypothesis, and trajectory ensembles.

## 5  Algorithmic Solutions

The generic SRM procedures described in Section 4 can lead to the design of a range of different algorithmic solutions for forecasting time series, depending on the choice of the families $\mathcal{H}_k$ and $\mathcal{F}_k$. The key challenge for the design of an algorithm design in this setting is to come up with a tractable procedure for searching through sets of increasing complexity. In this section, we describe one such procedure that leads to a boosting-style algorithm. Our algorithm learns a structural time series model by adaptively adding various structural subcomponents to the model in order to balance model complexity and the ability of the model to handle non-stationarity in data. We refer to our algorithm as *Boosted Structural Time Series Models* (BOOSTSM).

We will discuss BOOSTSM in the context of the squared loss, but most of our results can be straightforwardly extended to other convex loss functions. The hypothesis set used by our algorithm admits the following form: $\mathcal{H} = \{(\mathbf{x}, \mathbf{s}) \mapsto \mathbf{w} \cdot \Psi(\mathbf{x}) + \mathbf{w}' \cdot \mathbf{s} \colon \|\mathbf{w}\|_1 \leq \Lambda, \|\mathbf{w}'\|_1 \leq \Lambda'\}$. Each coordinate $\Psi_j$ is a binary-valued decision tree maps its inputs to a bounded set. For simplicity, we also assume that $\Lambda = \Lambda' = 1$. We choose $\mathcal{G}$ to be any set of state trajectories. For instance, this set may include periodic or trend sequences as described in Section 4.

Note that, to make the discussion concrete, we impose an $\ell_1$-constraint to the parameter vectors, but other regularization penalties are also possible. The particular choice of the regularization defined by $\mathcal{H}$ would also lead to sparser solutions, which is an additional advantage given that our state-space representation is high-dimensional.

For the squared loss and the aforementioned $\mathcal{H}$, the optimistic SRM objective (9) is given by

$$F(\mathbf{w}, \mathbf{w}') = \frac{1}{T} \sum_{t=1}^{T} \left( y_t - \mathbf{w} \cdot \Psi(\mathbf{x}_t) + \mathbf{w}' \cdot \mathbf{s}_t \right)^2 + \lambda(\|\mathbf{w}\|_1 + \|\mathbf{w}'\|_1), \tag{10}$$

where we omit $\log(k)$ because the index $k$ in our setting tracks the maximal depth of the tree and it suffices to restrict the search to the case $k < T$ as, for deeper trees, we can achieve zero empirical error. With this upper bound on $k$, $O\left(\sqrt{\frac{\log T}{T}}\right)$ is small and hence not included in the objective.

```
BOOSTSM(S = ((x_i, y_i)_{t=1}^T)
 1   f_0 ← 0
 2   for k ← 1 to K do
 3        j ← argmin_j ε_{k,j} + λ sgn(w_j)
 4        j' ← argmin_{j'} δ_{k,j'} + λ sgn(w'_j)
 5        if ε_{k,j} + λ sgn(w_j) ≤ δ_{k,j'} + λ sgn(w'_j) then
 6             η_k ← argmin_η F(w + ηε_j, w')
 7             f_k ← f_{k−1} + η_k Ψ_j
 8        else  η_k ← argmin_η F(w, w' + ηε_{j'})
 9             f_k ← f_{k−1} + η_t ε_{j'}
10   return f_K
```

Figure 1: Pseudocode of the BOOSTSM algorithm. On line 3 and 4 two candidates are selected to be added to the ensemble: a state trajectory with $j'$ or a tree-based predictor with index $j$. Both of these minimize their subgradients within their family of weak learners. Subgradients are defined by (11). The candidate with the smallest gradient is added to the ensemble. The weight of the new ensemble member is found via line search (line 6 and 8).

The regularization penalty is directly derived from the bounds on the expected sequential covering numbers of $\mathcal{H}$ given in Appendix B in Lemma 4 and Lemma 5.

Observe that (10) is a convex objective function. Our BOOSTSM algorithm is defined by the application of coordinate descent to this objective. Figure 1 gives its pseudocode. The algorithm proceeds in $K$ rounds. At each round, we either add a new predictor tree or a new state-space trajectory to the model, depending on which results in a greater decrease in the objective. In particular, with the following definitions:

$$\epsilon_{k,j} = \frac{1}{T} \sum_{t=1}^T (y_t - f_{k-1}(\mathbf{x}_t, \mathbf{s}_t))\Psi_j(\mathbf{x}_t), \quad \delta_{k,j} = \frac{1}{T} \sum_{t=1}^T (y_t - f_{k-1}(\mathbf{x}_t, \mathbf{s}_t))s_{t,j}. \quad (11)$$

the subgradient in tree-space direction $j$ at round $k$ is given by $\epsilon_{k,j} + \lambda \operatorname{sgn}(w_{k,j})$. We use the notation $\mathbf{w}_k$ to denote the tree-space parameter vector after $k-1$ rounds. Similarly, the subgradient in the trajectory space direction $j'$ is given by $\delta_{k,j'} + \lambda \operatorname{sgn}(w'_{k,j})$, where $\mathbf{w}'_k$ represents the trajectory space parameter vector after $k-1$ rounds.

By standard results in optimization theory [Luo and Tseng, 1992], BOOSTSM admits a linear convergence guarantee.

# 6   Conclusion

We introduced a new family of models for forecasting non-stationary time series, *Discriminative State-Space Models*. This family includes existing generative models and other state-based discriminative models (e.g. RNNs) as special cases, but also covers several novel algorithmic solutions explored in this paper. We presented an analysis of the problem of learning DSSMs in the most general setting of non-stationary stochastic processes and proved finite-sample data-dependent generalization bounds. These learning guarantees are novel even for traditional state-space models since the existing guarantees are only asymptotic and require the model to be correctly specified. We fully analyzed the complexity of several DSSMs that are useful in practice. Finally, we also studied the generalization guarantees of several structural risk minimization approaches to this problem and provided an efficient implementation of one such algorithm which is based on a convex objective. We report some promising preliminary experimental results in Appendix D.

**Acknowledgments**

This work was partly funded by NSF CCF-1535987 and NSF IIS-1618662, as well as a Google Research Award.

## Footnotes

[1]A more general formulation is given in terms of distribution of $Y_t$: $p_h(Y_t|S_t)p_g(S_t|S_{t-1})$.

[2]An alternative performance metric commonly considered in the time series literature is the *averaged* generalization error $\mathcal{L}_{T+1}(f) = \mathbb{E}[L(f, s_{T+1}, Z_{T+1})]$. The path-dependent generalization error that we consider in this work is a finer measure of performance since it only takes into consideration the realized history of the stochastic process, as opposed to an average trajectory.

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
