[Supplementary Material · dssm_supp.pdf]

# A Proofs

One of the key quantities that is used in our proofs is the empirical process:

$$\Phi(h,g) = \frac{1}{T}\sum_{t=1}^{T}\mathcal{L}_t(h,g|\mathbf{Z}_1^{t-1}) - \frac{1}{T}\sum_{t=1}^{T}L(h,s_t,Z_t), \tag{12}$$

where $(h,g)$ is an element of $\mathcal{F}$ and $\mathbf{s}$ the sequence generated by $g$. The following result gives a high-probability bound on the supremum of such empirical process.

Another definition that is required in the proofs below is the notion of decoupled tangent sequence. Given a sequence of random variables $\mathbf{Z}_1^T$ we say that $\mathbf{Z}'^T_1$ is a decoupled tangent sequence if $Z'_t$ is distributed according to $\mathbb{P}(\cdot|\mathbf{Z}_1^{t-1})$ and is independent of $\mathbf{Z}_t^\infty$. It is always possible to construct such a sequence of random variables [De la Peña and Giné, 1999]. We begin with the following concentration result.

**Lemma 1.** *Fix $\epsilon > 2\alpha > 0$. Then, the following holds:*

$$\mathbb{P}\left(\sup_{h,g\in\mathcal{F}}\Phi(h,g)\geq\epsilon\right) \leq \mathop{\mathbb{E}}_{\mathbf{v}\sim T(\mathbf{p})}\left[\mathcal{N}_1(\alpha,\mathcal{R},\mathbf{v})\right]\exp\left(-\frac{T(\epsilon-2\alpha)^2}{2M^2}\right).$$

*Symmetric result with $\Phi(h,g)$ replaced by $-\Phi(h,g)$ also holds.*

*Proof.* Note that, for any sequence $z_1,\ldots,z_T$, we can write sequence $L(h,s_1,z_1),\ldots,L(h,s_T,z_T)$, (where $\mathbf{s}$ is generated by $g$) as a sequence $f_1(z_1),\ldots,f_T(z_T)$. The rest of the proof follows the same arguments as in Theorem 1 in Kuznetsov and Mohri [2015] where instead of a fixed $f$ at each time $t$, we use $f_t$. For completeness, we include the full derivation of the result.

By Markov's inequality, for any $\lambda > 0$, the following inequality holds:

$$\mathbb{P}\left(\sup_{f\in\mathcal{F}}\left(\frac{1}{T}\sum_{t=1}^{T}(\mathbb{E}[f_t(Z_t)|\mathbf{Z}_1^{t-1}] - f_t(Z_t))\right)\geq\epsilon\right)$$

$$\leq \exp(-\lambda\epsilon)\,\mathbb{E}\left[\exp\left(\lambda\sup_{f\in\mathcal{F}}\left(\frac{1}{T}\sum_{t=1}^{T}(\mathbb{E}[f_t(Z_t)|\mathbf{Z}_1^{t-1}] - f_t(Z_t))\right)\right)\right].$$

If $\mathbf{Z}'^T_1$ is a tangent sequence the following equalities hold: $\mathbb{E}[f_t(Z_t)|\mathbf{Z}_1^{t-1}] = \mathbb{E}[f_t(Z'_t)|\mathbf{Z}_1^{t-1}] = \mathbb{E}[f_t(Z'_t)|\mathbf{Z}_1^{T}]$. Using these equalities and Jensen's inequality, we obtain the following:

$$\mathbb{E}\left[\exp\left(\lambda\sup_{f\in\mathcal{F}}\sum_{t=1}^{T}\frac{1}{T}\big(\mathbb{E}[f_t(Z_t)|\mathbf{Z}_1^{t-1}] - f_t(Z_t)\big)\right)\right]$$

$$= \mathbb{E}\left[\exp\left(\lambda\sup_{f\in\mathcal{F}}\mathbb{E}\Big[\sum_{t=1}^{T}\frac{1}{T}\big(f_t(Z'_t) - f_t(Z_t)\big)|\mathbf{Z}_1^{T}\Big]\right)\right]$$

$$\leq \mathbb{E}\left[\exp\left(\lambda\sup_{f\in\mathcal{F}}\frac{1}{T}\sum_{t=1}^{T}\big(f_t(Z'_t) - f_t(Z_t)\big)\right)\right],$$

where the last expectation is taken over the joint measure of $\mathbf{Z}_1^T$ and $\mathbf{Z}'^T_1$. Applying Lemma 2, we can further bound this expectation by

$$\mathop{\mathbb{E}}_{(\mathbf{z},\mathbf{z}')\sim T(\mathbf{p})}\mathop{\mathbb{E}}_{\sigma}\left[\exp\left(\lambda\sup_{f\in\mathcal{F}}\frac{1}{T}\sum_{t=1}^{T}\sigma_t\Big(f_t(\mathbf{z}'_t(\boldsymbol{\sigma})) - f_t(\mathbf{z}_t(\boldsymbol{\sigma}))\Big)\right)\right]$$

$$\leq \mathop{\mathbb{E}}_{(\mathbf{z},\mathbf{z}')\sim T(\mathbf{p})}\mathop{\mathbb{E}}_{\sigma}\left[\exp\left(\lambda\sup_{f\in\mathcal{F}}\frac{1}{T}\sum_{t=1}^{T}\sigma_t f_t(\mathbf{z}'_t(\boldsymbol{\sigma})) + \lambda\sup_{f\in\mathcal{F}}\sum_{t=1}^{T}-\sigma_t f_t(\mathbf{z}_t(\boldsymbol{\sigma}))\right)\right]$$

$$\leq \tfrac{1}{2}\mathop{\mathbb{E}}_{(\mathbf{z},\mathbf{z}')}\mathop{\mathbb{E}}_{\sigma}\left[\exp\left(2\lambda\sup_{f\in\mathcal{F}}\frac{1}{T}\sum_{t=1}^{T}\sigma_t f_t(\mathbf{z}'_t(\boldsymbol{\sigma}))\right)\right] + \tfrac{1}{2}\mathop{\mathbb{E}}_{(\mathbf{z},\mathbf{z}')}\mathop{\mathbb{E}}_{\sigma}\left[\exp\left(2\lambda\sup_{f\in\mathcal{F}}\frac{1}{T}\sum_{t=1}^{T}\sigma_t f_t(\mathbf{z}_t(\boldsymbol{\sigma}))\right)\right]$$

$$= \mathop{\mathbb{E}}_{\mathbf{z}\sim T(\mathbf{p})}\mathop{\mathbb{E}}_{\sigma}\left[\exp\left(2\lambda\sup_{f\in\mathcal{F}}\frac{1}{T}\sum_{t=1}^{T}\sigma_t f_t(\mathbf{z}_t(\boldsymbol{\sigma}))\right)\right],$$

where for the second inequality we used Young's inequality and for the last equality we used symmetry. Given $\mathbf{z}$ let $C$ denote the minimal $\alpha$-cover with respect to the $\ell_1$-norm of $\mathcal{F}$ on $\mathbf{z}$. Then, the following bound holds

$$\sup_{f \in \mathcal{F}} \frac{1}{T} \sum_{t=1}^{T} \sigma_t f_t(\mathbf{z}_t(\boldsymbol{\sigma})) \leq \max_{\mathbf{c} \in C} \frac{1}{T} \sum_{t=1}^{T} \sigma_t \mathbf{c}_t(\boldsymbol{\sigma}) + \alpha.$$

By the monotonicity of the exponential function,

$$\mathbb{E}_{\sigma} \left[ \exp \left( 2\lambda \sup_{f \in \mathcal{F}} \frac{1}{T} \sum_{t=1}^{T} \sigma_t f_t(\mathbf{z}_t(\boldsymbol{\sigma})) \right) \right] \leq \exp(2\lambda\alpha) \mathbb{E}_{\sigma} \left[ \exp \left( 2\lambda \max_{\mathbf{c} \in C} \frac{1}{T} \sum_{t=1}^{T} \sigma_t \mathbf{c}_t(\boldsymbol{\sigma}) \right) \right]$$

$$\leq \exp(2\lambda\alpha) \sum_{\mathbf{c} \in C} \mathbb{E}_{\sigma} \left[ \exp \left( 2\lambda \sum_{t=1}^{T} \sigma_t q_t \mathbf{c}_t(\boldsymbol{\sigma}) \right) \right].$$

Since $\mathbf{c}_t(\boldsymbol{\sigma})$ depends only on $\sigma_1, \ldots, \sigma_{T-1}$, by Hoeffding's bound,

$$\mathbb{E}_{\sigma} \left[ \exp \left( 2\lambda \frac{1}{T} \sum_{t=1}^{T} \sigma_t \mathbf{c}_t(\boldsymbol{\sigma}) \right) \right] = \mathbb{E} \left[ \exp \left( 2\lambda \frac{1}{T} \sum_{t=1}^{T-1} \sigma_t \mathbf{c}_t(\boldsymbol{\sigma}) \right) \mathbb{E}_{\sigma_T} \left[ \exp \left( 2\lambda \sigma_T \frac{1}{T} \mathbf{c}_T(\boldsymbol{\sigma}) \right) \Big| \boldsymbol{\sigma}_1^{T-1} \right] \right]$$

$$\leq \mathbb{E} \left[ \exp \left( 2\lambda \frac{1}{T} \sum_{t=1}^{T-1} \sigma_t \mathbf{c}_t(\boldsymbol{\sigma}) \right) \exp(2\lambda^2 \frac{1}{T^2} M^2) \right]$$

and iterating this inequality and using the union bound, we obtain the following:

$$\mathbb{P} \left( \sup_{f \in \mathcal{F}} \frac{1}{T} \sum_{t=1}^{T} (\mathbb{E}[f(Z_t)|\mathbf{Z}_1^{t-1}] - f(Z_t)) \geq \epsilon \right) \leq \mathbb{E}_{\mathbf{v} \sim T(\mathbf{p})} [\mathcal{N}_1(\alpha, \mathcal{R}, \mathbf{v})] \exp \left( -\lambda(\epsilon - 2\alpha) + 2\frac{1}{T}\lambda^2 M^2 \right).$$

Optimizing over $\lambda$ completes the proof of the first statement. The second statement holds by symmetry. $\square$

The following lemma is used in the proof of Lemma 1.

**Lemma 2.** *Given a sequence of random variables $\mathbf{Z}_1^T$ with joint distribution $\mathbf{p}$, let $\mathbf{Z'}_1^T$ be a decoupled tangent sequence. Then, for any measurable function $G$, the following equality holds*

$$\mathbb{E} \left[ G \Big( \sup_{f \in \mathcal{F}} \frac{1}{T} \sum_{t=1}^{T} (f_t(Z_t') - f_t(Z_t)) \Big) \right] = \mathbb{E}_{\boldsymbol{\sigma}} \mathbb{E}_{\mathbf{z} \sim T(\mathbf{p})} \left[ G \Big( \sup_{f} \frac{1}{T} \sum_{t=1}^{T} \sigma_t (f_t(\mathbf{z}_t'(\boldsymbol{\sigma})) - f_t(\mathbf{z}_t(\boldsymbol{\sigma}))) \Big) \right]. \tag{13}$$

*Proof.* The proof follows an argument in the proof of Theorem 3 of [Rakhlin et al., 2011]. We only need to check that every step holds for a time-varying $f_t$ instead $f$ fixed over time and for an arbitrary measurable function $G$, instead of the identity function. Observe that we can write the left-hand side of (13) as

$$\mathbb{E} \left[ G \Big( \sup_{f \in \mathcal{F}} \Sigma(\boldsymbol{\sigma}) \Big) \right] = \mathbb{E}_{Z_1, Z_1' \sim \mathbf{p}_1} \mathbb{E}_{Z_2, Z_2' \sim \mathbf{p}_2(\cdot | Z_1)} \cdots \mathbb{E}_{Z_T, Z_T' \sim \mathbf{p}_T(\cdot | \mathbf{Z}_1^{T-1})} \left[ G \Big( \sup_{f \in \mathcal{F}} \Sigma(\boldsymbol{\sigma}) \Big) \right],$$

where $\boldsymbol{\sigma} = (1, \ldots, 1) \in \{\pm 1\}^T$ and $\Sigma(\boldsymbol{\sigma}) = \sum_{t=1}^{T} \sigma_t \frac{1}{T} (f_t(Z_t') - f_t(Z_t))$. Now, by definition of decoupled tangent sequences, the value of the last expression is unchanged if we swap the sign of any $\sigma_{i-1}$ to $-1$ since that is equivalent to permuting $Z_i$ and $Z_i'$. Thus, the last expression is in fact equal to

$$\mathbb{E}_{Z_1, Z_1' \sim \mathbf{p}_1} \mathbb{E}_{Z_2, Z_2' \sim \mathbf{p}_2(\cdot | S_1(\sigma_1))} \cdots \mathbb{E}_{Z_T, Z_T' \sim \mathbf{p}_T(\cdot | S_1(\sigma_1), \ldots, S_{T-1}(\sigma_{T-1}))} \left[ G \Big( \sup_{f \in \mathcal{F}} \Sigma(\boldsymbol{\sigma}) \Big) \right]$$

for any sequence $\boldsymbol{\sigma} \in \{\pm 1\}^T$, where $S_t(1) = Z_t$ and $Z_t'$ otherwise. Since this equality holds for any $\boldsymbol{\sigma}$, it also holds for the mean with respect to uniformly distributed $\boldsymbol{\sigma}$. Therefore, the last expression is equal to

$$\mathbb{E}_{\boldsymbol{\sigma}} \mathbb{E}_{Z_1, Z_1' \sim \mathbf{p}_1} \mathbb{E}_{Z_2, Z_2' \sim \mathbf{p}_2(\cdot | S_1(\sigma_1))} \cdots \mathbb{E}_{Z_T, Z_T' \sim \mathbf{p}_T(\cdot | S_1(\sigma_1), \ldots, S_{T-1}(\sigma_{T-1}))} \left[ G \Big( \sup_{f \in \mathcal{F}} \Sigma(\boldsymbol{\sigma}) \Big) \right].$$

This last expectation coincides with the expectation with respect to drawing a random tree $\mathbf{z}$ from $T(\mathbf{p})$ (and its tangent tree $\mathbf{z}'$) and a random path $\boldsymbol{\sigma}$ to follow in that tree. That is, the last expectation is equal to

$$\mathop{\mathbb{E}}_{\boldsymbol{\sigma}} \mathop{\mathbb{E}}_{\mathbf{z} \sim T(\mathbf{p})} \left[ G\left( \sup_f \frac{1}{T} \sum_{t=1}^{T} \sigma_t(f_t(\mathbf{z}'_t(\boldsymbol{\sigma})) - f_t(\mathbf{z}_t(\boldsymbol{\sigma}))) \right) \right],$$

which concludes the proof. $\qquad\square$

Now we proceed with the proofs of our main results.

**Theorem 1.** *Fix* $\mathbf{s} \in \mathcal{S}^{T+1}$. *For any* $\delta > 0$, *with probability at least* $1 - \delta$, *for all* $h \in \mathcal{H}$ *and all* $\alpha > 0$,

$$\mathcal{L}(f|\mathbf{Z}_1^T) \leq \frac{1}{T} \sum_{t=1}^{T} L(h, X_t, s_t) + \mathrm{disc}(\mathbf{s}) + 2\alpha + M\sqrt{\frac{2 \log \frac{\mathbb{E}_{\mathbf{v} \sim T(\mathbb{P})}[\mathcal{N}_1(\alpha, \mathcal{R}_\mathbf{s}, \mathbf{v})]}{\delta}}{T}},$$

*where* $\mathcal{R}_\mathbf{s} = \{(z, s) \mapsto L(h, s, z) \colon h \in \mathcal{H}\} \times \{\mathbf{s}\}$.

*Proof.* The proof of this result immediately follows from Theorem 2 by fixing $\mathcal{G}$ to be a singelton $\{g\}$, where $g$ generates $\mathbf{s}$ on any input. $\qquad\square$

**Theorem 2.** *For any* $\delta > 0$, *with probability at least* $1 - \delta$, *for all* $f = (h, g) \in \mathcal{H} \times \mathcal{G}$ *and all* $\alpha > 0$,

$$\mathcal{L}(f|\mathbf{Z}_1^T) \leq \frac{1}{T} \sum_{t=1}^{T} L(h, X_t, s_t) + \mathrm{disc}(\mathbf{s}) + 2\alpha + M\sqrt{\frac{2 \log \frac{\mathbb{E}_{\mathbf{v} \sim T(\mathbb{P})}[\mathcal{N}_1(\alpha, \mathcal{R}, \mathbf{v})]}{\delta}}{T}},$$

*where* $s_t = g(X_t, s_{t-1})$ *for all* $t$ *and* $\mathcal{R} = \{(z, s) \mapsto L(h, s, z) \colon h \in \mathcal{H}\} \times \mathcal{G}$.

*Proof.* Define the following empirical process

$$\Phi_0(h, g) = \mathcal{L}(f|\mathbf{Z}_1^T) - \frac{1}{T} \sum_{t=1}^{T} L(h, X_t, s_t).$$

Observe that since difference of supermums is upper bounded by the supremum of the difference the following inequality holds

$$\sup_{g \in \mathcal{G}} \left( \sup_{h \in \mathcal{H}} \Phi_0(h, g) - \mathrm{disc}(\mathbf{s}_g) \right) \leq \sup_{g \in \mathcal{G}} \sup_{h \in \mathcal{H}} \Phi(h, g),$$

where we use notation $\mathbf{s}_g$ to emphasize dependence of state sequences $\mathbf{s}$ on $g$. The result now follows by application of Lemma 1. $\qquad\square$

**Theorem 3.** *For any* $\delta > 0$, *with probability at least* $1 - \delta$, *for all* $\alpha > 0$, *the following bound holds:*

$$\mathcal{L}_{T+1}(\widetilde{h}, \widetilde{g}|\mathbf{Z}_1^T) \leq \mathcal{L}_{T+1}(f^*|\mathbf{Z}_1^T) + 2\Delta(\mathbf{s}^*) + 2\alpha + 2B_{k(f^*)} + M\sqrt{\frac{\log k(f^*)}{T}} + 2M\sqrt{\frac{\log \frac{2}{\delta}}{T}},$$

*where* $s_t^* = g^*(X_t, s_{t-1}^*)$ *and* $k(f^*)$ *is the smallest integer* $k$ *such that* $f^* \in \mathcal{H}_k \times \mathcal{G}_k$.

*Proof.* To simplify the notation, we let $\mathcal{N}(k) = \mathbb{E}_{\mathbf{v} \sim T(\mathbb{P})}[\mathcal{N}_1(\alpha, \mathcal{R}_k, \mathbf{v})]$. Observe that the following chain of inequalities holds:

$$\mathbb{P}\left( \sup_{k \geq 1} \sup_{f \in \mathcal{F}_k} \mathcal{L}_{T+1}(f|\mathbf{Z}_1^T) - F(f, k) - 2\alpha \geq \epsilon \right)$$

$$\leq \sum_{k=1}^{\infty} \mathbb{P}\left( \sup_{f \in \mathcal{F}_k} \mathcal{L}_{T+1}(f|\mathbf{Z}_1^T) - F(f, k) - 2\alpha \geq \epsilon \right)$$

$$= \sum_{k=1}^{\infty} \mathbb{P}\left( \sup_{g \in \mathcal{G}_k} \Phi_0(h, g) - \text{disc}(\mathbf{s}) \geq \epsilon + 2\alpha + M\sqrt{\frac{\log k}{T}} + M\sqrt{\frac{2 \log \mathcal{N}(k)}{T}} \right)$$

$$\leq \sum_{k=1}^{\infty} \mathcal{N}(k) \exp\left( -\frac{1}{2M^2} T\left( \epsilon + M\sqrt{\frac{\log k}{T}} + M\sqrt{\frac{2 \log \mathcal{N}(k)}{T}} \right)^2 \right)$$

$$\leq \sum_{k=1}^{\infty} \mathcal{N}(k) \exp\left( -\frac{1}{2M^2} T\epsilon^2 - 2 \log k - \log \mathcal{N}(k) \right)$$

$$= \exp\left( -\frac{2T\epsilon^2}{M^2} \right) \sum_{k=1}^{\infty} \frac{1}{k^2}$$

$$= \frac{\pi^2}{6} \exp\left( -\frac{2T\epsilon^2}{M^2} \right)$$

$$\leq 2 \exp\left( -\frac{2T\epsilon^2}{M^2} \right),$$

where we use the union bound for the first inequality, Theorem 2 for the second inequality and $(a + b)^2 \leq 2a^2 + 2b^2$ for the third inequality. Next, we observe that

$$\mathbb{P}\left( \mathcal{L}_{T+1}(\widetilde{h}, \widetilde{g}|\mathbf{Z}_1^T) - \mathcal{L}_{T+1}(f^*|\mathbf{Z}_1^T) - 2\Delta(\mathbf{s}^*) - 2\alpha - 2B_{k(f^*)} - M\sqrt{\frac{\log k(f^*)}{T}} > \epsilon \right)$$

$$\leq \mathbb{P}\left( \mathcal{L}_{T+1}(\widetilde{h}, \widetilde{g}|\mathbf{Z}_1^T) - F(\widetilde{f}, \widetilde{k}) - \alpha \geq \frac{\epsilon}{2} \right)$$

$$+ \mathbb{P}\left( F(f^*, k(f^*)) - \mathcal{L}_{T+1}(f^*|\mathbf{Z}_1^T) - 2\Delta(\mathbf{s}^*) - \alpha - 2B_{k(f^*)} - M\sqrt{\frac{\log k(f^*)}{T}} > \frac{\epsilon}{2} \right),$$

where we used the union bound and the fact that $F(\widetilde{f}, \widetilde{k}) - F(f^*, k(f^*)) \leq 0$ since $\widetilde{f}$ is minimizer of $F$. The first term is bounded by $2 \exp\left( -\frac{2T\epsilon^2}{M^2} \right)$ by the previous argument. Similarly, by Lemma 1, the second term is bounded by $2 \exp\left( -\frac{2T\epsilon^2}{M^2} \right)$ since

$$F(f^*, k(f^*)) - \mathcal{L}_{T+1}(f^*|\mathbf{Z}_1^T) - 2\Delta(\mathbf{s}^*) - \alpha - 2B_{k(f^*)} - M\sqrt{\frac{\log k(f^*)}{T}}$$

can be upper bounded by

$$\frac{1}{T} \sum_{t=1}^{T} L(h, s_t, Z_t) - \mathcal{L}_{T+1}(f^*|\mathbf{Z}_1^T) - \alpha - \text{disc}(\mathbf{s}^*) - M\sqrt{\frac{2 \log \mathcal{N}(k(f^*))}{T}}.$$

This completes the proof of this result. □

**Theorem 4.** *Under Assumption 1, for any $\delta > 0$, with probability at least $1 - \delta$, for all $\alpha > 0$,*
$\mathcal{L}_{T+1}(h_0, g_0|\mathbf{Z}_1^T) - \mathcal{L}_{T+1}(f^*|\mathbf{Z}_1^T) < r(\epsilon)$, *where*

$$\epsilon = 2\alpha + 2B_{k(f^*)} + M\sqrt{\frac{\log k(f^*)}{T}} + 2M\sqrt{\frac{\log \frac{2}{\delta}}{T}},$$

*where $s_t^* = g^*(X_t, s_{t-1}^*)$ and $k(f^*)$ is the smallest integer $k$ such that $f^* \in \mathcal{H}_k \times \mathcal{G}_k$.*

*Proof.* Observe that the following decomposition holds

$$\frac{1}{T}\sum_{t=1}^{T}\mathcal{L}_t(h_0,g_0|\mathbf{Z}_1^{t-1}) - \mathcal{L}_t(h^*,g^*|\mathbf{Z}_1^{t-1}) = \left(\frac{1}{T}\sum_{t=1}^{T}\mathcal{L}_t(h_0,g_0|\mathbf{Z}_1^{t-1}) - \widetilde{F}(h_0,g_0,k_0)\right)$$
$$+\left(\widetilde{F}(h_0,g_0,k_0) - \widetilde{F}(h^*,g^*,k^*)\right)$$
$$+\left(\widetilde{F}(h^*,g^*,k^*) - \frac{1}{T}\sum_{t=1}^{T}\mathcal{L}_t(h^*,g^*|\mathbf{Z}_1^{t-1})\right).$$

Since $\widetilde{F}(h_0,g_0,k_0) - \widetilde{F}(h^*,g^*,k^*) \leq 0$ by definition of $\widetilde{F}$ and $(h_0,g_0,k_0)$, it follows from the union bound that

$$\mathbb{P}\left(\frac{1}{T}\sum_{t=1}^{T}\mathcal{L}_t(h_0,g_0|\mathbf{Z}_1^{t-1}) - \mathcal{L}_t(h^*,g^*|\mathbf{Z}_1^{t-1}) - 2\alpha - 2B_{k(f^*)} - M\sqrt{\frac{\log k(f^*)}{T}} > \epsilon\right)$$

$$\leq \mathbb{P}\left(\frac{1}{T}\sum_{t=1}^{T}\mathcal{L}_t(h_0,g_0|\mathbf{Z}_1^{t-1}) - \widetilde{F}(h_0,g_0,k_0) - \alpha \geq \frac{\epsilon}{2}\right)$$

$$+\mathbb{P}\left(\widetilde{F}(h^*,g^*,k^*) - \frac{1}{T}\sum_{t=1}^{T}\mathcal{L}_t(h^*,g^*|\mathbf{Z}_1^{t-1}) - \alpha - 2B_{k(f^*)} - M\sqrt{\frac{\log k(f^*)}{T}} > \frac{\epsilon}{2}\right).$$

Since we can upper bound

$$\widetilde{F}(h^*,g^*,k^*) - \frac{1}{T}\sum_{t=1}^{T}\mathcal{L}_t(h^*,g^*|\mathbf{Z}_1^{t-1}) - \alpha - 2B_{k(f^*)} - M\sqrt{\frac{\log k(f^*)}{T}}$$

$$\leq -\Phi(h^*,g^*) - \alpha - M\sqrt{\frac{2\log\mathcal{N}(k(f^*))}{T}},$$

the second term is bounded by $2\exp\left(-\frac{2T\epsilon^2}{M^2}\right)$. We are using notation $\mathcal{N}(k) = \mathbb{E}_{\mathbf{v}\sim T(\mathbb{P})}[\mathcal{N}_1(\alpha,\mathcal{R}_k,\mathbf{v})]$ to simplify the presentation. To complete the proof, observe that the following chain of inequalities holds, by union bound and Lemma 1:

$$\mathbb{P}\left(\sup_{k\geq 1}\sup_{f\in\mathcal{F}_k}\frac{1}{T}\sum_{t=1}^{T}\mathcal{L}_t(f|\mathbf{Z}_1^{t-1}) - \widetilde{F}(f,k) - 2\alpha \geq \epsilon\right)$$

$$\leq \sum_{k=1}^{\infty}\mathbb{P}\left(\sup_{f\in\mathcal{F}_k}\frac{1}{T}\sum_{t=1}^{T}\mathcal{L}_t(f|\mathbf{Z}_1^{t-1}) - \widetilde{F}(f,k) - 2\alpha \geq \epsilon\right)$$

$$= \sum_{k=1}^{\infty}\mathbb{P}\left(\sup_{(h,g)\in\mathcal{F}}\Phi(h,g) \geq \epsilon + 2\alpha + M\sqrt{\frac{\log k}{T}} + M\sqrt{\frac{2\log\mathcal{N}(k)}{T}}\right)$$

$$\leq \sum_{k=1}^{\infty}\mathcal{N}(k)\exp\left(-\frac{1}{2M^2}T\left(\epsilon + M\sqrt{\frac{\log k}{T}} + M\sqrt{\frac{2\log\mathcal{N}(k)}{T}}\right)^2\right).$$

Arguments similar to the ones in the proof of Theorem 3 show that this term also bounded $2\exp\left(-\frac{2T\epsilon^2}{M^2}\right)$. Therefore, by Assumption 1 it follows that for any $\delta > 0$, with probability at least $1-\delta$, for all $\alpha > 0$, $\mathcal{L}_{T+1}(h_0,g_0|\mathbf{Z}_1^T) - \mathcal{L}_{T+1}(f^*|\mathbf{Z}_1^T) < r(\epsilon)$ and this completes the proof. $\square$

# B  Complexities of DSSMs

In this Section, we prove upper bounds on the expected sequential covering numbers of some commonly used hypothesis sets. We start with a following lemma that gives a general upper bound in terms of the sequential Rademacher complexity:

$$\mathfrak{R}_T^{\text{seq}}(L\circ\mathcal{H}) = \sup_{\mathbf{z},\mathbf{s}}\mathbb{E}_{\epsilon,\sigma}\left[\sup_{h\in\mathcal{H}}\left(\frac{1}{T}\sum_{t=1}^{T}\sigma_t L(h,\mathbf{s}_t(\sigma),z_t(\sigma))\right)\right], \tag{14}$$

Table 1: Weighted sequential cover and sequential covering numbers.

where the supremum is taken over all pairs of complete $\mathcal{Z}$-valued binary tree $\mathbf{z}$ and $\mathcal{S}$-valued binary tree $\mathbf{s}$.

**Lemma 3.** *The following bound holds:*

$$\sup_{\alpha > 0} \frac{\alpha}{2} \sqrt{\log \mathcal{N}_2(2\alpha, \mathcal{R})} \leq 3\sqrt{\frac{\pi}{2} \log T} \, \mathfrak{R}_T^{seq}(L \circ \mathcal{H}),$$

*whenever $\mathcal{N}_2(2\alpha, \mathcal{R}) < \infty$.*

*Proof.* As in the previous proofs, we use the convention that for any sequence of $z_1, \ldots, z_T$, we can write sequence $L(h, s_1, z_1), \ldots, L(h, s_T, z_T)$, (where $\mathbf{s}$ is generated by $g$) as a sequence $f_1(z_1), \ldots, f_T(z_T)$. We consider the following Gaussian-Rademacher sequential complexity:

$$\mathfrak{G}_T^{\text{seq}}(\mathcal{F}, \mathbf{z}) = \mathbb{E}_{\gamma, \sigma} \left[ \sup_{f \in \mathcal{F}} \left( \frac{1}{T} \sum_{t=1}^{T} \sigma_t \gamma_t f_t(z_t(\sigma)) \right) \right], \tag{15}$$

where $\sigma$ is an independent sequence of Rademacher random variables, $\gamma$ is an independent sequence of standard Gaussian random variables and $\mathbf{z}$ is a complete binary tree of depth $T$ with values in $\mathcal{Z}$.

Observe that if $V$ is any $\alpha$-cover with respect to the $\ell_2$-norm of $\mathcal{F}$ on $\mathbf{z}$, then, the following holds by independence of $\gamma$ and $\sigma$:

$$\mathfrak{G}^{\text{seq}}(\mathcal{F}, \mathbf{z}) \geq \mathbb{E}_{\gamma} \mathbb{E}_{\sigma} \left[ \sup_{\mathbf{v} \in V} \left( \frac{1}{T} \sum_{t=1}^{T} \sigma_t \gamma_t \mathbf{v}_t(\sigma) \right) \right] = \mathbb{E}_{\sigma} \mathbb{E}_{\gamma} \left[ \sup_{\mathbf{v} \in V} \left( \frac{1}{T} \sum_{t=1}^{T} \sigma_t \gamma_t \mathbf{v}_t(\sigma) \right) \right].$$

Notice that $V$ is also a $2\alpha$-cover with respect to the $\ell_2$-norm of $\mathcal{F}$ on $\mathbf{z}$. We can obtain a smaller $2\alpha$-cover $V_0$ from $V$ by eliminating $\mathbf{v}$s that are $\alpha$-close to some other $\mathbf{v}' \in V$. Since $V$ is finite, let $V = \{\mathbf{v}^1, \ldots, \mathbf{v}^{|V|}\}$, and for each $\mathbf{v}^i$ we delete $\mathbf{v}^j \in \{\mathbf{v}_{i+1}, \ldots, \mathbf{v}_{|V|}\}$ such that the following inequality holds:

$$\left[ \frac{1}{T} \sum_{t=1}^{T} \left( \mathbf{v}_t^i(\sigma) - \mathbf{v}_t^j(\sigma) \right)^2 \right]^{1/2} \leq \alpha.$$

It is straightforward to verify that $V_0$ is a $2\alpha$-cover with respect to the $\ell_2$-norm of $\mathcal{F}$ on $\mathbf{z}$. Furthermore, it follows that for a fixed $\sigma$, the following holds:

$$\mathbb{E}_{\gamma} \left[ \left( \frac{1}{T} \sum_{t=1}^{T} \sigma_t \gamma_t \mathbf{v}_t(\sigma) - \sum_{t=1}^{T} \sigma_t \gamma_t \mathbf{v}'_t(\sigma) \right)^2 \right] \geq \alpha^2$$

for any $\mathbf{v}', \mathbf{v} \in V_0$. Let $Z_i, i = 1, \ldots, |V_0|$ be a sequence of independent Gaussian random variables with $\mathbb{E}[Z_i] = 0$ and $\mathbb{E}[Z_i^2] = \alpha^2/2$. Observe that $\mathbb{E}[(Z_i - Z_j)] = \alpha^2$ and hence by the Sudakov-Fernique inequality it follows that

$$
\begin{aligned}
\mathbb{E}_{\boldsymbol{\sigma}} \mathbb{E}_{\boldsymbol{\gamma}} \left[ \sup_{\mathbf{v} \in V} \left( \frac{1}{T} \sum_{t=1}^{T} \sigma_t \gamma_t \mathbf{v}_t(\boldsymbol{\sigma}) \right) \right] &\geq \mathbb{E}_{\boldsymbol{\sigma}} \mathbb{E}_{\boldsymbol{\gamma}} \left[ \sup_{\mathbf{v} \in V_0} \left( \frac{1}{T} \sum_{t=1}^{T} \sigma_t \gamma_t \mathbf{v}_t(\boldsymbol{\sigma}) \right) \right] \\
&\geq \mathbb{E} \left[ \max_{i=1,\ldots,|V_0|} Z_i \right] \\
&\geq \frac{\alpha}{2} \sqrt{\log |V_0|},
\end{aligned}
$$

where the last inequality is the standard result for Gaussian random variables. Therefore, we conclude that $\mathfrak{G}^{\mathrm{seq}}(\mathcal{F}, \mathbf{z}) \geq \sup_{\alpha>0} \frac{\alpha}{2} \sqrt{\log \mathcal{N}_2(2\alpha, \mathcal{F}, \mathbf{z})}$. On the other hand, using standard properties of Gaussian complexity Ledoux and Talagrand [1991], we have

$$
\mathfrak{G}_T^{\mathrm{seq}}(\mathcal{F}, \mathbf{z}) \leq 3 \sqrt{\frac{\pi}{2} \log T} \, \mathbb{E}_{\boldsymbol{\epsilon}, \boldsymbol{\sigma}} \left[ \sup_{f \in \mathcal{F}} \left( \frac{1}{T} \sum_{t=1}^{T} \sigma_t \epsilon_t f_t(z_t(\boldsymbol{\sigma})) \right) \right],
$$

where $\boldsymbol{\epsilon}$ is an independent sequence of Rademacher variables. We re-arrange $\mathbf{z}$ into $\mathbf{z}^{\boldsymbol{\epsilon}}$ so that $z_t(\boldsymbol{\sigma}) = z_t^{\boldsymbol{\epsilon}}(\boldsymbol{\epsilon}\boldsymbol{\sigma})$ for all $\boldsymbol{\sigma} \in \{\pm 1\}^T$ and it follows that

$$
\begin{aligned}
\mathbb{E}_{\boldsymbol{\epsilon}, \boldsymbol{\sigma}} \left[ \sup_{f \in \mathcal{F}} \left( \frac{1}{T} \sum_{t=1}^{T} \sigma_t \epsilon_t f_t(z_t(\boldsymbol{\sigma})) \right) \right] &= \mathbb{E}_{\boldsymbol{\epsilon}, \boldsymbol{\sigma}} \left[ \sup_{f \in \mathcal{F}} \left( \frac{1}{T} \sum_{t=1}^{T} \sigma_t \epsilon_t f_t(z_t^{\boldsymbol{\epsilon}}(\boldsymbol{\epsilon}\boldsymbol{\sigma})) \right) \right] \\
&\leq \mathbb{E}_{\boldsymbol{\epsilon}} \left[ \sup_{\mathbf{z}} \mathbb{E}_{\boldsymbol{\sigma}} \left[ \sup_{f \in \mathcal{F}} \left( \frac{1}{T} \sum_{t=1}^{T} \sigma_t \epsilon_t f_t(z_t(\boldsymbol{\epsilon}\boldsymbol{\sigma})) \right) \right] \right] \\
&= \sup_{\mathbf{z}} \mathbb{E}_{\boldsymbol{\sigma}} \left[ \sup_{f \in \mathcal{F}} \left( \frac{1}{T} \sum_{t=1}^{T} \sigma_t f_t(z_t(\boldsymbol{\sigma})) \right) \right] \\
&\leq \sup_{\mathbf{z}, \mathbf{s}} \mathbb{E}_{\boldsymbol{\epsilon}, \boldsymbol{\sigma}} \left[ \sup_{h \in \mathcal{H}} \left( \frac{1}{T} \sum_{t=1}^{T} \sigma_t L(h, \mathbf{s}_t(\boldsymbol{\sigma}), z_t(\boldsymbol{\sigma})) \right) \right].
\end{aligned}
$$

Therefore, the following inequality holds

$$
\sup_{\alpha>0} \frac{\alpha}{2} \sqrt{\log \mathcal{N}_2(2\alpha, \mathcal{F}, \mathbf{z})} \leq 3 \sqrt{\frac{\pi}{2} \log T} \, \mathfrak{R}_T^{\mathrm{seq}}(L \circ \mathcal{H}),
$$

and the conclusion of the theorem follows by taking the supremum with respect to $\mathbf{z}$ on both sides of this inequality. $\qquad\square$

The following lemma decomposes the complexity of structural time series models into the complexities of its subcomponents.

**Lemma 4.** *Let $p \geq 1$ and $L \circ \mathcal{H} = \{(\mathbf{x}, \mathbf{s}, y) \to (\mathbf{w} \cdot \Psi(\mathbf{x}) + \mathbf{w}' \cdot \mathbf{s} - y)^p \colon \mathbf{w} \in S, \mathbf{w}' \in S'\}$ for some $S, S'$. Assume that the condition $|\mathbf{w} \cdot \mathbf{x} - y| \leq M$ holds for all $(\mathbf{x}, y) \in \mathcal{Z}$ and all $\mathbf{w}$ such that $\|\mathbf{w}\|_{\mathcal{H}} \leq \Lambda$. Then, the following inequalities hold:*

$$
\mathfrak{R}_T^{seq}(\mathcal{F}) \leq pM^{p-1}C_T(\mathfrak{R}_T^{seq}(H_1) + \mathfrak{R}_T^{seq}(H_1)), \tag{16}
$$

*where $H_1 = \{\mathbf{x} \to \mathbf{w} \cdot \Psi(\mathbf{x}) \colon \mathbf{w} \in S\}$, $H_2 = \{\mathbf{s} \to \mathbf{w}' \cdot \mathbf{s} \colon \mathbf{w}' \in S'\}$ and $C_T = 8(1 + 4\sqrt{2} \log^{3/2}(eT^2))$.*

*Proof.* Since $x \to |x|^p$ is $pM^{p-1}$-Lipschitz over $[-M, M]$, by Lemma 13 in [Rakhlin et al., 2015a], the following bound holds:

$$
\mathfrak{R}_T^{\mathrm{seq}}(L \circ \mathcal{F}) \leq pM^{p-1}C_T \mathfrak{R}_T^{\mathrm{seq}}(H'),
$$

where $H' = \{(\mathbf{x}, \mathbf{s}, y) \to \mathbf{w} \cdot \Psi(\mathbf{x}) + \mathbf{w}' \cdot \mathbf{s} - y : \mathbf{w} \in S, \mathbf{w} \in S'\}$. Note that Lemma 13 requires that $\mathfrak{R}_T^{\text{seq}}(H') > 1/T$ which is guaranteed by Khintchine's inequality. By definition of the sequential Rademacher complexity

$$
\begin{aligned}
\mathfrak{R}_T^{\text{seq}}(H') &= \sup_{(\mathbf{x},y)} \mathbb{E}_{\boldsymbol{\sigma}}\left[ \sup_{\mathbf{w}} \frac{1}{T} \sum_{t=1}^{T} \sigma_t (\mathbf{w} \cdot \Psi(\mathbf{x}_t(\boldsymbol{\sigma})) - y(\boldsymbol{\sigma})) \right] \\
&= \sup_{\mathbf{x}} \mathbb{E}_{\boldsymbol{\sigma}}\left[ \sup_{\mathbf{w} \in S} \frac{1}{T} \sum_{t=1}^{T} \sigma_t \mathbf{w} \cdot \Psi(\mathbf{x}_t(\boldsymbol{\sigma})) \right] + \sup_{\mathbf{s}} \mathbb{E}_{\boldsymbol{\sigma}}\left[ \sup_{\mathbf{w}' \in S'} \frac{1}{T} \sum_{t=1}^{T} \sigma_t \mathbf{w}' \cdot \mathbf{s}_t(\boldsymbol{\sigma}) \right] \\
&\quad + \sup_{y} \mathbb{E}_{\boldsymbol{\sigma}}\left[ \frac{1}{T} \sum_{t=1}^{T} \sigma_t y(\boldsymbol{\sigma}) \right] = \mathfrak{R}_T^{\text{seq}}(H_1) + \mathfrak{R}_T^{\text{seq}}(H_2),
\end{aligned}
$$

where, for the last equality, we used the fact that $\sigma_t$s are mean zero random variables and $\sigma_t$ is independent of $y(\boldsymbol{\sigma}) = y(\sigma_1, \sigma_2, \dots, \sigma_{t-1})$. This completes the proof. $\qquad\square$

Observe that, for example, if $H_1 = \{\mathbf{x} \to \mathbf{w} \cdot \Psi(\mathbf{x}) \colon \|\mathbf{w}\|_H \leq \Lambda\}$ where $H$ is a Hilbert space with a corresponding feature map $\Psi \colon \mathcal{X} \to \mathcal{H}$ and PDS kernel $K$, then, by Lemma 6 in [Kuznetsov and Mohri, 2015], we have the inequality $\mathfrak{R}_T^{\text{seq}}(H_1) \leq \frac{\Lambda r}{\sqrt{T}}$, where $r = \sup_x K(x, x)$. Similarly, if $H_2 = \{\mathbf{s} \to \mathbf{w} \cdot \mathbf{s} \colon \|\mathbf{w}\|_2 \leq \Lambda_0\}$ then we have $\mathfrak{R}_T^{\text{seq}}(H_2) \leq \frac{\Lambda_0 r_0}{\sqrt{T}}$, with $r_0 = \sup_\mathbf{s} \|\mathbf{s}\|_2$. The next result gives a bound on the sequential complexity of a weighted combination of $N$ binary-valued functions $\mathcal{H}_1$. A common choice of such binary-valued functions are decision trees. Note that result is logarithmic in the number of such functions, which suggests using a large set of functions.

**Lemma 5.** *Let $H = \{\mathbf{x} \to (\mathbf{w} \cdot \Psi(\mathbf{x}) \colon \|\mathbf{w}\|_1 \leq \Lambda\}$ where, for each $j \in [1, N]$, $\Psi_j$ is a binary-valued function. Then, the following inequalities hold:*

$$
\mathfrak{R}_T^{seq}(H) \leq \sqrt{\frac{2\Lambda^2 \log 2N}{T}}. \tag{17}
$$

*Proof.* We observe that, by the definition of the dual now, the following equalities hold:

$$
\begin{aligned}
\sup_{\mathbf{x}} \mathbb{E}_{\boldsymbol{\sigma}}\left[ \sup_{\|\mathbf{w}\|_1 \leq \Lambda} \sum_{t=1}^{T} \sigma_t \mathbf{w} \cdot \Psi(\mathbf{x}_t(\boldsymbol{\sigma})) \right] &= \Lambda \sup_{\mathbf{x}} \mathbb{E}_{\boldsymbol{\sigma}} \left\| \sum_{t=1}^{T} \sigma_t \Psi(\mathbf{x}_t(\boldsymbol{\sigma})) \right\|_\infty \\
&= \Lambda \sup_{\mathbf{x}} \mathbb{E}_{\boldsymbol{\sigma}}\left[ \max_{j \in [1,N], s \in \{\pm 1\}} \sum_{t=1}^{T} \sigma_t \Psi_j(\mathbf{x}_t(\boldsymbol{\sigma})) \right].
\end{aligned}
$$

To bound the last quantity, we apply an argument similar to the one used in the proof of Massart's lemma. The key difference is that here $\mathbf{x}_t$ depends on $\sigma$, which requires a more careful analysis. Observe that, by the monotonicity of exp, the following upper bound holds, for any $u > 0$:

$$
\exp\left( u \, \mathbb{E}_{\boldsymbol{\sigma}}\left[ \max_{j \in [1,N], s \in \{\pm 1\}} \sum_{t=1}^{T} \sigma_t \Psi_j(\mathbf{x}_t(\boldsymbol{\sigma})) \right] \right) \leq 2 \sum_{j=1}^{N} \mathbb{E}_{\boldsymbol{\sigma}}\left[ \exp\left( u \sum_{t=1}^{T} \sigma_t \Psi_j(\mathbf{x}_t(\boldsymbol{\sigma})) \right) \right].
$$

Next, since $\sigma_T$ and $\mathbf{x}_T(\boldsymbol{\sigma}) = \mathbf{x}_T(\sigma_1, \dots, \sigma_{T-1})$ are independent, by Hoeffding's bound, we can write

$$
\begin{aligned}
\mathbb{E}_{\boldsymbol{\sigma}}\left[ \exp\left( u \sum_{t=1}^{T} \sigma_t \Psi_j(\mathbf{x}_t(\boldsymbol{\sigma})) \right) \right] &= \mathbb{E}_{\boldsymbol{\sigma}_1^{T-1}}\left[ \exp\left( u \sum_{t=1}^{T-1} \sigma_t \Psi_j(\mathbf{x}_t(\boldsymbol{\sigma})) \right) \mathbb{E}_{\boldsymbol{\sigma}_1^{T-1}}\left[ \exp(u\sigma_T \Psi_j(\mathbf{x}_T(\boldsymbol{\sigma}))) \right] \right] \\
&\leq e^{2u} \mathbb{E}_{\boldsymbol{\sigma}_1^{T-1}}\left[ \exp\left( u \sum_{t=1}^{T-1} \sigma_t \Psi_j(\mathbf{x}_t(\boldsymbol{\sigma})) \right) \right].
\end{aligned}
$$

Iterating this result over $t$ and optimizing over $u$ yields the desired upper bound. $\qquad\square$

## C State Discrepancy Estimation

We can decompose the discrepancy as follows:

$$
\text{disc}(\mathbf{s}) \le \sup_{h \in \mathcal{H}} \left( \frac{1}{\tau} \sum_{t=T-\tau+1}^{T} \mathcal{L}_t(h, g | \mathbf{Z}_1^{t-1}) - \frac{1}{T} \sum_{t=1}^{T} \mathcal{L}_t(h, g | \mathbf{Z}_1^{t-1}) \right) \tag{18}
$$

$$
+ \sup_{h \in \mathcal{H}} \left( \mathcal{L}_{T+1}(h, g | \mathbf{Z}_1^T) - \frac{1}{\tau} \sum_{t=T-\tau+1}^{T} \mathcal{L}_t(h, g | \mathbf{Z}_1^{t-1}) \right).
$$

We will assume that the second term, which we will denote by $\text{disc}_\tau$, is sufficiently small and will show that the first term can be estimated from data. Note that, in general, the requirement that $\text{disc}_\tau$ is small is necessary for learning Barve and Long [1996].

The following result shows that we can estimate the first term appearing in the upper bound (18) on $\text{disc}(\mathbf{s})$.

**Theorem 5.** *For any $\delta > 0$, with probability at least $1 - \delta$, the following holds for all $\alpha > 0$:*

$$
\text{disc}(\mathbf{s}) - \widehat{\text{disc}}(\mathbf{s}) \le 2\alpha + M \|\mathbf{u} - \mathbf{u}_\tau\|_2 \sqrt{2 \log \frac{\mathbb{E}_{\mathbf{z} \sim T(\mathbf{p})}[\mathcal{N}_1(\alpha, \mathcal{R}, \mathbf{z})]}{\delta}},
$$

*where $\mathbf{u}$ is the uniform distribution over the sample, $\mathbf{u}_\tau$ a uniform distribution over the last $s$ points, and*

$$
\widehat{\text{disc}}(\mathbf{s}) = \sup_{h \in \mathcal{H}} \left( \frac{1}{\tau} \sum_{t=T-\tau+1}^{T} L(h, s_t, Z_t) - \frac{1}{T} \sum_{t=1}^{T} L(h, s_t, Z_t) \right). \tag{19}
$$

*Proof.* We upper bound the difference of suprema by the supremum of the difference and then apply the same arguments as in the proof of Lemma 1, with the only difference that the examples are here weighted by $(1/T - 1/\tau)$. □

## D Experiments

In this section, we present the results of experiments with the BOOSTSM algorithm.

As subcomponents of our BOOSTSM algorithm, we used a linear trend model and an ensemble of decision trees as described in Section 5. As a baseline comparator, we used a regular STSM model with subcomponents that consists of a trend model and a linear model. Note that, since our comparator can only choose a single trend model, to make the comparison fair, we restricted our algorithm to choose a single trend model as well.

For our experiments we used web traffic data from Wikipedia articles. Each of these time series represents a number of daily views (integer) of a different Wikipedia article starting from January, 1st, 2012 up to March, 1st, 2017. This data is obtained via public pageviews API `https://wikitech.wikimedia.org/wiki/Analytics/PageviewAPI`. In our experiments, we used time series for nine articles that appear among top twenty articles in 2016 (in terms of number of total views): `Bernie Sanders`, `Batman vs Superman`, `Brownian motion`, `Donald Trump`, `Java`, `Isaac Newton`, `Janet Jackson`, `Merle Haggard`, `Main Page`, `Ted Cruz`.

The following experimental setup was used in our experiments. Each algorithm was trained on the first 1500 days, then used to predict the next 25 days. The average error was recorded and then the algorithm was retrained with 1525 days and used to predict the next 25 and so on. We report the average error across all these rounds in Table 2. We also report the running errors in Figure 3.

Since the time series used in our experiments have vastly different scales, both within time series and between different time series, RMSE is not an appropriate evaluation metric. Thus, for ease of comparison, we chose Symmetric Absolute Percentage Error (SMAPE) as our evaluation metric. This loss function is defined by $L(x, y) = |x - y| / (|x| + |y|)$ if at least one of $x, y$ is not zero and $L(x, y) = 0$ otherwise. Note, however, that both algorithms were trained using the squared loss, as discussed in the previous sections. SMAPE is not a convex loss function and is generally hard to

Table 2: Mean SMAPE for BOOSTSMand STSM.

| | BOOSTSM | STSM |
|---|---|---|
| Batman vs Superman | **0.6060 ± 0.0958** | 0.6489 ± 0.1056 |
| Bernie Sanders | 0.7247 ± 0.0805 | **0.7118 ± 0.0829** |
| Brownian motion | **0.4226 ± 0.0499** | 0.6078 ± 0.1118 |
| Donald Trump | **0.7124 ± 0.0727** | 0.7304 ± 0.0603 |
| Java | **0.5800 ± 0.1497** | 0.7982 ± 0.0451 |
| Isaac Newton | **0.4896 ± 0.0911** | 0.5988 ± 0.0823 |
| Janet Jackson | **0.7003 ± 0.0671** | 0.7150 ± 0.0565 |
| Merle Haggard | 0.7400 ± 0.0703 | **0.7205 ± 0.0728** |
| Main Page | **0.5639 ± 0.1041** | 0.5809 ± 0.0958 |
| Ted Cruz | **0.6910 ± 0.0972** | 0.7272 ± 0.1067 |

optimize, which is why we used MSE as our optimization objective. The ranking of the models in our experiments in terms of RMSE is the same as the obtained using SMAPE.

In our experiments, BOOSTSM outperformed STSM on eight out of ten datasets, in some cases by a substantial margin. This suggests that BOOSTSM may be able to better adapt to different types of non-stationarities present in the data.

Table 3: Running SMAPE for BOOSTSM (blue) and STSM (green). Datasets appear in alphabetic order: left to right, top to bottom.