[Reviews · NeurIPS 2017]

Reviewer 1



In _Discriminative State Space Models_, the authors build on ‘Discrepancy’ as introduced by Kuznetov and Mohri [2015] to prove generalization guarantees for state space models. In particular they give a Structural Risk Minimization set up with guarantees for state space models. A boosting-like algorithm is given for SRM framework. Besides, proofs very preliminary experiments are given in the supplementary materials. The primary contribution of this paper is theoretical. As a reviewer I have neither a background in sequential rademacher complexity nor the ’Discrepancy” of Kuznetov and Mohri, so I am unable to contact on the technical achievements of this papers. Part algorithmic/part theoretical is the SRM framework for DSSMs culminating in the boosting-like algorithm. Although bounds are given their is little other intuition or insight that I could find about the methodology within the body or in the supplementary material, e.g., What other approaches could be considered here? Should we suppose this SRM approach works well in practice? Minor: L95-106 : Sequential covering number extension to state space models was hard to follow, needed to go to Rahklin, 2010 to understand. Noting the that the depth of tree is T at the beginning definitions rather than at the end, would make it easier for a first reader. 164 upper -> upper bound

Reviewer 2



This paper studies a setting of non-stationary time series forecasting. They compare there performance to Discriminative State-Space Models though no assumption is made on the underlying stochastic process (mispecified model). There general analysis obtains finite time data-dependant guarantees. And there algorithm based on penalized risk minimization leads to several novel algorithms for forecasting time series. To my opinion, this paper gives a nice contribution to the litterature and captures several families of generative models as special cases. They also improve some existing result to finite-time guarantees with misspecified model. The paper is well-written though the notion of sequential covering is quite hard to understand. A figure would be helpfull. Some remarks: - It would be nice to write the complexity of you algorithmic solutions for the different examples in the main text. - Is Assumption 1 your only assumption on the loss? Don't you have some Lipschitz, convexity assumption? I was also wondering if you couldn't get an intermediary result on the cumulative risk without Assumption 1. - Do you have any lower-bound for the term disc(s)? - I was a bit surprised of Th3: k is calibrated automatically though you are dealing with an infinite number of arbitrary families R_k? - Why in the experiments, do you test and train with different loss functions? Local remarks: - l111: Z_2' - Th1: - l130: "if distribution" -> "if the distribution" - l271: "an algorithm design" - l287: I hardly see the connexion with Eq (9). What about B_k? Why does it suffices to consider $k < T$? - Algo: pb with parenthesis. - The figure p20 are hard to read in B&W: can you use dashed/dotted lines?

Reviewer 3



# Summary The authors build on and generalize the work of Kuznetsov and Mohri [2015], to come up with generalization bounds for learning a large class of state-space models. Based on these bounds, a structural risk minimization formulation is proposed to estimate forecasting models with learning guarantees both in the case where the state-space predictor is not neccesarily accurate and in the case where we assume that it is. The authors show that for many models of interest, this is a reasonable assumption. The convex objective function of the resulting SRM is then solved using a coordinate descent algorithm, with some encouraging empirical results presented in the appendix. # Remarks Thank you for presenting a dense but very interesting paper. Unfortunately, I have few suggestions or insights to offer for improving the technique itself, but I will attempt to offer some suggestions that would help me better understand and appreciate the presented material. In line 75-76 you mention that the loss function is bounded by M, yet in the next line you consider the squared loss. Is there a restriction on x,y or h that I am missing that guarantees M < L? I do not follow the argument in lines 201-206: the paragraph seems to claim that SRM is useful for finding the right capacity of the state predictor, for instance, to make sure assumption 1 holds, and defends this by referring to theorem 4, which itself is based on assumption 1. Perhaps this paragraph could be clarified to avoid this seemingly circular reasoning. You then go on to prove for various interesting models that assumption 1 does indeed hold. It may be useful to stress that all but the last example consider realizable cases (if I understand them correctly), because this will have some consequences as to which bound you would want to use (Theorem 3 vs. Theorem 4). You use the word "separable" on two occasions, but it's meaning is not clear to me from the context. I was a confused by the model in section 5, where the class of state predictors is restricted to a singleton. Does this not mean that we effectively do not have to learn the states (making it less remarkable that the objective is convex)? And is this not a such simpler setting than the more general scenario describes in the rest of the paper where G (and H) can be a richer class of functions? In line 291 it is not clear to me from which specific result in appendix B this claim follows. Finally, after finishing the paper the reason for the title was still unclear to me. Why are they called discriminative? Especially since, as you mention, they cover a number of models that are traditionally considered generative models? Some typos I came across: 130: if distribution -> if the distribution 143: The "is" is in the wrong place 145: a uniformly -> uniformly 260: underlying -> the underlying 302: enjoys -> enjoys a # Author Comments Thank you for your comments. While I understand the issue surrounding the G as a singleton in Sec. 5 a bit better now, I hope you can spend some more time explaining this in a revised version, for now it is unclear to me what the effect on the bounds is of adding a large number of state trajectories in this singleton and in what circumstances we would want G to be something else than a singleton.

Reviewer 4



Synopsis ------------------- * This paper follows in a line of work from Kuznetsov and Mohri on developing agnostic learning guarantees for time series prediction using sequential complexity measures. Ostensibly, the setup in past works and the present work is that one will fit a hypothesis to observations (x_t,y_t) from time 1 through T, then use this hypothesis to predict y_{T+1} from x_{T+1} (the generalization guarantees provided are slightly more general than this). Where previous works (e.g. "Learning Theory and Algorithms for Forecasting Non-Stationary Time Series", NIPS 2015) considered the case where the hypothesis class is a fixed mapping from context space to outcome space, the present work extends this setup to the case where the setting where one learns a "state space" model. Here, the each hypothesis is a pair (h,g), where g generates an internal "state" variable and h predicts using both a context and state variable (see the paper for more detail). This captures, for example, HMMs. The authors give basic generalization guarantees for this setup, generalization for the setup under structural risk minimization, and an algorithm for learning a restricted class of state space models. Review ------------------- * I found the concept for the paper exciting and believe it is well-written but I feel it has two major shortcomings: I) Technical novelty * The main results Theorem 1 and 2 should be compared with Theorem 1/Corollary 2 of the NIPS 2015 paper mentioned above (call it KM15 for short). The authors mention that the main advantage of the present work is that the discrepancy term in Theorem 2 does not include a sup over the class G, which one would get by applying Theorem 1 from KM15 naively. Yet, if we look at the proof of Theorem 1 Theorem 2 of the present work, we see that this is achieved by being careful with where one takes the sup over G (line 418), then immediately applying to Lemma 1, whose proof is essentially identical to that of KM15 Theorem 1. To give some more examples: Appendix B largely consists of standard facts about sequential rademacher complexity. It is unclear to me whether the structural risk minimization results represent a substantive improvement on the basic generalization bounds. While I admit that others may find these results practically useful, I feel that the applications and experiments should have been fleshed out further if this was the goal. II) General setting is not instantiated sufficiently. * I was not satisfied with the depth to which the general bounds (eg Theorem 2) were instantiated. I do not feel that the basic promise of the paper (learning state space models) has been fully explored. Namely ** The algorithms section (5) only applies in the setting where the state space mapping class G is a singleton, and so there is no real "learning" of the state space dynamics going on. While I acknowledge that developing efficient algorithms for learning G may be challenging, this bothers me from a conceptual perspective because Theorem 2 bears no improvement over the KM15 results mentioned above when G only contains a single element. ** In line with this comment, the discrepancy term never seems to be instantiated. ** While Theorem 2 depends on the expected sequential cover of the class H x G, which one expects should exhibit some rich dependence on the dynamics G and the distribution over (X,Y), Appendix B seems to handle this term by immediately bounding it by the *worst case* sequential rademacher complexity, which includes a sup over all state space paths, not just those that could be induced by G. Conclusion ------------------- * In summary, I found the basic concept fairly interesting and think that the tools introduced (eg sequential covering for state space models) will be useful, but I don't think they were explored to enough depth. I also found Assumption 1 and the concrete cases where the discrepancy can be removed in the SRM setting to form a nice set of observations. * One last comment: While you allude that some notion of discrepancy is necessary for learning based on some results of Long, it would be nice to see a formal lower bound whose functional form depends on this quantity in the general case. Misc comments: * Line 123: Definition of \mathcal{R}_{s} doesn't exactly match the formal specification for the class \mathcal{R} used in the definition of the sequential covering number.